# CoastalBench: A Decade-Long High-Resolution Dataset to Emulate Complex Coastal Processes

**Zelin Xu** [1]  **Yupu Zhang** [1]  **Tingsong Xiao** [1]  **Maitane Olabarrieta** [2]  **Jose M. Gonzalez-Ondina** [2]  **Zibo Liu** [1]
**Shigang Chen** [1]  **Zhe Jiang** [1]

## Abstract

Over 40% of the global population lives within 100 kilometers of the coast, which contributes more than $8 trillion annually to the global economy. Unfortunately, coastal ecosystems are increasingly vulnerable to more frequent and intense extreme weather events and rising sea levels. Coastal scientists use numerical models to simulate complex physical processes, but these models are often slow and expensive. In recent years, deep learning has become a promising alternative to reduce the cost of numerical models. However, progress has been hindered by the lack of a large-scale, high-resolution coastal simulation dataset to train and validate deep learning models. Existing studies often focus on relatively small datasets and simple processes. To fill this gap, we introduce a decade-long, high-resolution ($<$100m) coastal circulation modeling dataset on a real-world 3D mesh in southwest Florida with around 6 million cells. The dataset contains key oceanography variables (e.g., current velocities, free surface level, temperature, salinity) alongside external atmospheric and river forcings. We evaluated a customized Vision Transformer model that takes initial and boundary conditions and external forcings and predicts ocean variables at varying lead times. The dataset provides an opportunity to benchmark novel deep learning models for high-resolution coastal simulations (e.g., physics-informed machine learning, neural operator learning). The code and dataset can be accessed at https://github.com/spatialdatasciencegroup/CoastalBench.

---

[1]Department of Computer & Information Science & Engineering, University of Florida, Gainesville, FL, USA [2]Department of Civil & Coastal Engineering, University of Florida, Gainesville, FL, USA. Correspondence to: Zhe Jiang <zhe.jiang@ufl.edu>.

*Proceedings of the 42$^{st}$ International Conference on Machine Learning*, Vancouver, Canada. PMLR 267, 2025. Copyright 2025 by the author(s).

## 1. Introduction

Over 40% of the global population lives within 100 kilometers of the coast, with these coastal regions contributing more than $8 trillion annually to the global economy (Martínez et al., 2007; Nicholls, 2004). Coastal ecosystems provide a multitude of services that support human populations and natural habitats, such as regulating the global climate, protecting against natural disasters, and sustaining food security through fisheries (Barbier et al., 2011). Unfortunately, these ecosystems are increasingly vulnerable to more frequent and intense extreme weather events, rising sea levels, and additional stressors such as habitat degradation, invasive species, and pollution (He & Silliman, 2019). Coastal scientists develop numeric models to simulate complex coastal processes such as wave propagation, tidal dynamics, wind-driven circulation, temperature dynamics, and salinity dynamics in estuaries (Neumann et al., 2017). Unfortunately, such numerical models are computationally expensive. Recently, deep learning has provided a promising alternative to emulate complex coastal processes with neural networks, significantly reducing the computational costs (James et al., 2018; Wei & Davison, 2022).

However, progress has been hindered by the lack of a large-scale, high-resolution coastal dataset to train and validate deep learning models. Effectively modeling coastal processes requires a high spatial (e.g., 100m) and temporal (e.g., 30 minutes) resolution. The spatial resolution often needs to reach even a few meters near inlets in estuaries to capture complex land-water interactions.

As summarized in Table 1, existing studies on deep learning for coastal processes primarily focus on relatively small datasets for specific coastal processes, such as wave propagation (James et al., 2018; Wei & Davison, 2022; Xu et al., 2025), sea level rise (Nieves et al., 2021; Ishida et al., 2020), and sea surface temperature (Chen et al., 2024; Song et al., 2024; Nieves et al., 2021). There are also works on training physics-informed neural networks (PINNs) (Raissi et al., 2019) for coastal flood and storm surge modeling (Feng et al., 2023; Chen et al., 2022). However, training a PINN in real-world complex topography is very challenging, if

*Table 1.* Comparison of existing datasets used for deep learning for coastal processes.

| Dataset & Reference | Spatial Resolution | Temporal Resolution | Time Span | Variables |
|---|---|---|---|---|
| (O'Donncha et al., 2019) | $\sim$1 km | 1 hour | 4 years (2013-2017) | Wave condition, wind speed |
| (Jiang et al., 2021) | 7 km | 5 min | 1 year (2020) | Wind speed, pressure, sea surface level |
| (de Melo et al., 2023) | $\sim$1–10 m | 10 min | A few days | Wave condition, wind speed |
| (Kumar & Leonardi, 2023) | $\sim$1 km | - | 500 storm simulations | Wave condition, sediment transport |
| (Shen et al., 2024) | $\sim$100 m | 1 day | 5 years (1991 - 1995) | Wave condition, wind speed |
| (Wu et al., 2024) | $\sim$25 km | 1 month | 29 years (1993-2021) | Sea surface air temperature, salinity, wind speed, Chlorophyll-a concentration, atmospheric $pCO_2$, etc |
| **CoastalBench (Ours)** | $\sim$100 m | 30 min | 10 years (2008–2017) | Wave condition, temperature, salinity, sea surface air pressure, air temperature, humidity, rainfall, sun radiation, wind speed, etc. |

possible. Another relevant body of research is deep learning for global weather forecasting (Lam et al., 2023; Bi et al., 2023; Nguyen et al., 2024; Hersbach et al., 2020; Rasp et al., 2024). The main difference is that these works focus on low-resolution atmospheric simulations on a global scale, while we focus on high-resolution regional simulations.

To fill these gaps, this paper introduces *CoastalBench*, a decade-long (2008–2017), high-resolution coastal simulation dataset generated using the Regional Ocean Modeling System (ROMS) (Shchepetkin & McWilliams, 2005). Covering an estuary near Charlotte Harbor, Florida, USA, the dataset features a 3D mesh with over 6 million cells, an average horizontal grid resolution of less than 100m, and a 30-minute temporal resolution. It includes simulated ocean variables (e.g., current velocity, salinity, temperature, free surface elevation), external forcings (e.g., meteorological forcings, river forcings), and static physical features (e.g., bathymetry). *CoastalBench* contains physical variables related to multiple processes, such as wave propagation, tidal dynamics, wind-driven circulation, temperature dynamics, and salinity dynamics in estuaries. For example, storm surge and coastal flood forecasting benefit from predictions of free surface elevation; water quality and stratification modeling rely on temperature and salinity; and sediment transport analysis depends on vertical diffusivity.

We further evaluate a customized Vision Transformer (ViT) model (Dosovitskiy et al., 2020) for this dataset, which encodes input boundary conditions into conditional attention operated on patches of initial conditions and meteorological forcings. The model can predict coastal variables at varying lead times. In summary, the contributions of this paper are as follows:

- We release *CoastalBench*, a decade-long, high-resolution coastal circulation simulation dataset generated from the ROMS model. To the best of our knowledge, this is the first open dataset to capture complex coastal processes at such a high spatiotemporal resolution and scale, providing a unique opportunity to benchmark novel deep-learning models in coastal oceanography.

- We design a straightforward yet effective ViT-based model for regional coastal processes by leveraging the dataset and incorporating static features, external forcings, initial and boundary conditions, and other key factors.

- Through experimental results, we demonstrate the promising performance of the deep learning model by comparing it with ROMS simulations. An ablation study validates the importance of components, while a scaling test analyzes the impact of model size on predictive accuracy.

## 2. Preliminaries

The coastal ocean is a highly dynamic environment, influenced by the complex interplay of atmospheric forcing, terrestrial runoff, oceanic boundary inputs, and bathymetry. This work formulates the problem of learning a deep learning model to emulate ROMS simulations of coastal dynamics, given appropriate physical inputs. The model is conditioned on key components that govern the ocean state evolution, including initial and boundary conditions, external forcings, and static spatial features. We begin by introducing relevant terminologies used throughout this work.

**Definition 2.1. Initial Conditions (IC)** specify the three-dimensional ocean state at the start of the simulation ($t = t_0$). These include fields such as velocity, temperature, and salinity, denoted as $\mathbf{X} \in \mathbb{R}^{C \times H \times W \times D}$, where $C$ is the number of variables, and $H, W, D$ are the height, width, and depth of the 3D study area, respectively. Formally, the initial conditions can be expressed as:

$$\mathbf{X}_{\text{IC}} = \mathbf{X}(t_0), \qquad (1)$$

**Definition 2.2. Boundary Conditions (BC)** specify the values of the state variables, $\mathbf{X}$, at the lateral edges of the simulation domain, denoted as $\partial\Omega$. These conditions represent interactions with adjacent open ocean regions and are typically prescribed based on larger-scale circulation models or observational datasets. For our purposes, we focus on lateral boundary conditions:

$$\mathbf{X}_{\text{BC}}(t) = \mathbf{X}(t)\big|_{\partial\Omega} \tag{2}$$

**Definition 2.3. External Forcings** are time-varying drivers that influence the internal evolution of the coastal ocean system. These include: **Meteorological Forcings**, including surface fields such as wind stress, air temperature, pressure, humidity, shortwave/longwave radiation, and precipitation, typically derived from meteorological reanalysis; **River Inflow**, including freshwater inflows from rivers that influence coastal stratification, salinity, and circulation.

Then, our problem can be formally defined as follows:

**Input:**
- Initial conditions $\mathbf{X}_{\text{IC}} \in \mathbb{R}^{C \times H \times W}$, the state of variables at $t_0$.
- Lead time $\Delta t \in \mathbb{R}^+$, the time interval for prediction relative to $t_0$.
- Boundary conditions $\mathbf{X}_{\text{BC}}(t_0 + \Delta t) \in \mathbb{R}^{C \times H_{\text{lat}} \times W_{\text{lat}}}$, the state of variables at the lateral boundaries at $t_0 + \Delta t$.
- Meteorological forcings $\mathbf{f}_{\text{Meteo}}(t_0 + \Delta t) \in \mathbb{R}^{M \times H \times W}$, where $M$ is the number of meteorological variables.
- River inflow $\mathbf{f}_{\text{River}}(t_0 + \Delta t) \in \mathbb{R}^{N \times R}$, where $N$ is the number of river variables and $R$ is the number of rivers.
- Static time-invariant physical features $\mathbf{s} \in \mathbb{R}^{S \times H \times W}$, where $S$ is the number of features.

**Output:**
- Predicted variables $\hat{\mathbf{X}}(t_0 + \Delta t) \in \mathbb{R}^{C \times H \times W}$ at $t_0 + \Delta t$.

**Objective:**
- Minimize the prediction error of the deep learning model.

## 3. *CoastalBench*

We release *CoastalBench*, a decade-long, high-resolution dataset designed for modeling complex coastal processes. Table 2 provides an overview of CoastalBench. *CoastalBench* integrates diverse categories of variables, covering a large spatiotemporal domain with fine resolution. This dataset serves as a benchmark for developing and evaluating deep learning models for coastal process emulation, physics-informed machine learning, neural operator learning, and spatiotemporal forecasting.

The dataset is generated using the Regional Ocean Modeling System (ROMS), a widely used three-dimensional hydrostatic ocean circulation model designed for high-resolution simulations of coastal and regional ocean dynamics. It

*Table 2.* Overview of *CoastalBench*.

| Property | Description |
|---|---|
| Temporal Coverage | 2008–2017 (10 years) |
| Temporal Resolution | 30 minutes |
| Spatial Domain | Charlotte Harbor, Florida, USA ($\sim$800 km$^2$) |
| Spatial Resolution | Varying, $\sim$120 m $\times$ 100 m |
| Grid Type | Non-uniform 3D mesh |
| Grid Dimensions | $898 \times 598 \times 12$ |
| Number of Variables | 22 |
| Data Format | NetCDF |
| Total Volume | $\sim$18 TB |
| Simulation Model | Regional Ocean Modeling System (ROMS) |

solves the Reynolds-Averaged Navier–Stokes (RANS) equations under the hydrostatic and Boussinesq approximations, employing a terrain-following vertical coordinate system to accurately represent complex bathymetry and coastal processes (Haidvogel et al., 2008; Shchepetkin & McWilliams, 2005). ROMS incorporates advanced numerical schemes for advection and mixing, along with sophisticated turbulence closure models, making it well-suited for studying ocean circulation, estuarine dynamics, and ecosystem interactions. The model supports various physical parameterizations, data assimilation techniques, and multi-scale nesting capabilities, enabling its application across a wide range of oceanographic and coastal studies.

*Table 3.* Summary of physical variables included in *CoastalBench*. The dataset provides a comprehensive representation of ocean state variables, external forcings, and static spatial features, supporting machine learning models for coastal process modeling.

| Category | Variables |
|---|---|
| Ocean variables | Current velocity, water temperature, salinity, free surface elevation |
| Meteorological forcings | Air pressure, air temperature, air heat flux, precipitation, wind speed, radiation |
| River forcings | River runoff mass transport vertical profile, temperature, salinity |
| Static features | Bathymetry, Coriolis parameter, curvilinear coordinate metric |

**Variables.** As summarized in Table 3, *CoastalBench* includes key ocean state variables representing the evolving physical conditions of the coastal system. Additionally, it incorporates external forcings, such as meteorological forcings and river inflows, which drive regional spatiotemporal variability. Static spatial features, including bathymetry and

grid size, are also provided to capture the fixed geographical properties that influence coastal dynamics. These components together form a comprehensive dataset suitable for studying and modeling coastal ocean processes. Additional details on these variables can be found in Appendix A.

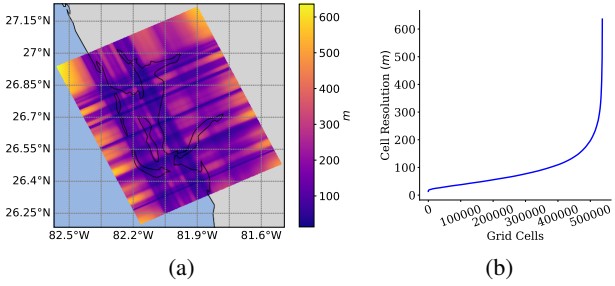

*Figure 1.* (a) Grid resolution computed as the square root of the area of grid cells (near Charlotte Harbor and Fort Myers). (b) The cumulative distribution of cell count at increasing resolutions.

**Spatiotemporal Coverage and Resolution.** *CoastalBench* provides high-frequency simulations with a temporal resolution of 30 minutes from 2008 to 2017. The dataset covers the Charlotte Harbor region, an area characterized by dynamic interactions between the atmosphere, land, and open ocean, spanning approximately 800 square kilometers. As shown in Figure 1, the model employs a horizontally non-uniform grid with varying resolution, ranging from ∼300 m offshore to ∼1 m in estuarine regions and inlets, with an average grid resolution of 88 m. The finest resolution occurs in shallow and estuarine regions because complex land-sea interactions require higher accuracy. Figure 1b presents the distribution of grid resolutions. The majority of cells have resolutions below 200 m, with only a small fraction exceeding 500 m. The vertical domain consists of 12 terrain-following layers, providing a resolution of ∼0.1 m in the shallowest regions of the estuary.

**Dataset Construction.** The dataset is generated using ROMS, which simulates ocean dynamics based on initial conditions, boundary conditions, and external forcings. The simulation is run in hindcast mode, using high-quality reanalysis data calibrated against historical observations. ROMS is forced at the ocean boundaries with free surface elevation, velocity, water temperature, and salinity obtained from the HYbrid Coordinate Ocean Model with the Naval Research Lab's Coupled Ocean Data Assimilation Global analysis. These forcings have a spatial resolution of 1/12 degree and a temporal resolution of 3 hours. For boundary conditions, the Chapman implicit condition is applied to free surface elevation, the Flather condition to depth-averaged currents, and radiation boundary conditions to temperature and salinity. Vertical mixing is parameterized using the Generic Length Scale turbulence closure scheme (Warner et al., 2005). Atmospheric forcings including 10 m wind

speed, surface pressure, air temperature, humidity, and radiation fluxes are obtained from the North American Regional Reanalysis (Mesinger et al., 2006), with a temporal resolution of 3 hours and a spatial resolution of 0.3 degrees. Freshwater inflows from the three main rivers in the modeled area (Myakka River, Peace River, and Caloosahatchee River) are acquired from US Geological Survey flow data[1]. Bathymetric data are sourced from NOAA's Continuously Updated Digital Elevation Model at 1/9 arc-second resolution and the US Geological Survey's 2018 Lidar Digital Elevation Model (0.5 m horizontal resolution). Before generating the final dataset, the model was calibrated through multiple short-term runs (∼months), with model parameters adjusted using observational data from the River, Estuary, and Coastal Observing Network (RECON)[2]. It is noted that although RECON provides observational data, its sparse spatial coverage and irregular sampling make high-resolution simulations necessary for comprehensive spatiotemporal modeling.

To ensure compatibility with deep learning models, post-processing is applied to the simulation outputs. Since certain ocean variables are defined on different grid locations (e.g., free surface elevation is located at the center of each cell, whereas velocity components are located at the cell boundaries), they are regridded onto the same coordinate system (center of grid cells). Additionally, atmospheric forcings, which originally have coarser spatiotemporal resolution, are resampled to match the resolution of the simulated ocean variables. After processing, the dataset is structured as a grid of size $898 \times 598 \times 12$ (height × width × depth) for each snapshot.

**Highlights.** The main highlights of *CoastalBench* are as follows:

- **High Resolution.** *CoastalBench* provides high-resolution real-world simulation data with a spatial resolution of <100 m, and a temporal resolution of 30 minutes, enabling the study of regional coastal ocean dynamics.

- **Large Scale.** The dataset spans a decade, covering approximately 800 $km^2$, with over 100,000 time steps and 6 million cells per snapshot.

- **Unique Structure.** Unlike traditional gridded datasets, *CoastalBench* is based on a non-uniform 3D mesh, which can be used for evaluating deep learning models designed for irregular spatial structures.

- **Complex Processes.** The dataset contains multiple physical processes with all necessary variables, providing a strong benchmark for evaluating machine learning methods for modeling physical processes.

---

[1]https://dashboard.waterdata.usgs.gov/
[2]https://recon.sccf.org/

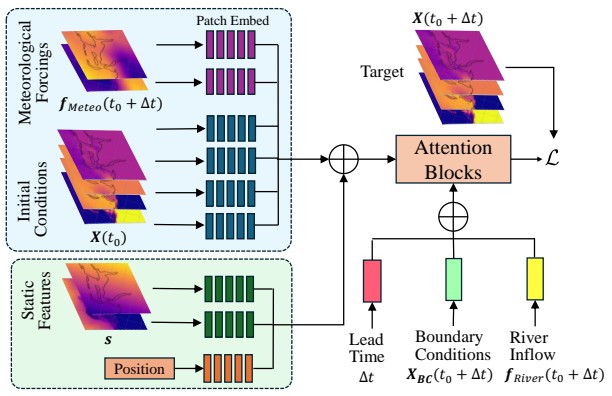

Figure 2. Overview of the model architecture.

# 4. Approach

As shown in Figure 2, we aim to train a deep learning model that predicts ocean variables at specific lead times, given the initial conditions, boundary conditions, external forcings, and static physical features. We design a simple ViT-based model and treat 3D variables at different depths as separate channels, flattening the 3D spatial structure into multiple 2D variables.

## 4.1. Model Architecture

We propose a simple ViT-based model customized for emulating regional coastal processes. First, instead of the standard patch embedding in ViT, which struggles to capture vertical relationships among variables and fails to model the impacts of meteorological forcings, we tokenize each coastal ocean variable and meteorological variable independently and aggregate information across them. Second, to address the limitations of standard positional embedding, we incorporate spatially varying but temporally invariant physical features (e.g., bathymetry, grid size) into the positional embedding, enabling the model to understand critical geographical properties. Finally, the model performs attention conditioned on lead time, boundary conditions, and river inflows. Conditioning on lead time allows predictions across multiple time intervals with a single model, while boundary conditions and river inflows capture key interactions with the external environment, which significantly influence the internal processes of the coastal system.

**Patch embedding of initial conditions and meteorological forcings.** Standard patch embedding operation in ViT, designed for image data, has limitations for our task: (1) it fails to model the vertical relationships between variables originally at different depths, and (2) tokenizing coastal ocean variables only ignores the impacts of external meteorological forcings. To address these issues, inspired by (Nguyen et al., 2023; 2024), we propose a customized patch

embedding. Each variable $\mathbf{x}_i \in \mathbb{R}^{H \times W}$, whether ocean or meteorological, is independently tokenized to a representation $\mathbf{z}_i \in \mathbb{R}^{\frac{H}{p} \times \frac{W}{p} \times d}$, where $p$ is the patch size and $d$ is the embedding dimension. Additionally, we inject a learnable vector $\mathbf{v}_i \in \mathbb{R}^d$ for each variable to encode the identity of each variable, which is added to its token embeddings. This yields an intermediate tensor of shape $\frac{H}{p} \times \frac{W}{p} \times (C+M) \times d$, where $C$ and $M$ are the numbers of ocean and meteorological variables, respectively. To aggregate information across variables, we apply a single-layer cross-attention mechanism over the variable dimension, using a learnable query to aggregate the embeddings. The resulting output, denoted as $\mathbf{z}_{agg} \in \mathbb{R}^{\frac{H}{p} \times \frac{W}{p} \times d}$, serves as the final patch embedding fed into the following attention blocks. Our patch embedding module is specifically designed to capture the spatiotemporal complexity of regional coastal processes by integrating meteorological forcings alongside ocean variables and modeling their interactions within each patch.

**Physics-aware positional embedding.** Although the standard positional embedding in ViTs can represent the relative positions of image patches, it does not understand the geographical and physical properties critical to modeling coastal processes. To address this, we incorporate static physical features $\mathbf{s} \in \mathbb{R}^{S \times H \times W}$, where $S$ is the number of physical variables (e.g., ocean depth, grid size, Coriolis parameter), into the positional embedding. These physical features encode essential geophysical information about a specific regional coastal system, enabling the model to better understand the underlying spatiotemporal dynamics. We formalize the physics-aware positional embedding as follows. Let $\mathbf{p} \in \mathbb{R}^{\frac{H}{p} \times \frac{W}{p} \times d}$ denote the standard positional embedding. The static physical features $\mathbf{s}$ are first patchified into embeddings $\mathbf{z}_{phy} \in \mathbb{R}^{\frac{H}{p} \times \frac{W}{p} \times d}$ using standard patch embedding operation. Then we use a cross-attention mechanism to allow the physical embeddings $\mathbf{z}_{phy}$ to interact with the standard positional embedding $\mathbf{p}$:

$$\mathbf{p}_{phy} = \text{CrossAttention}(\mathbf{z}_{phy}, \mathbf{p}, \mathbf{p}), \quad (3)$$

This approach ensures that the positional embedding reflects both the spatial layout of the patches and the geophysical properties of the coastal system.

**Conditional Attention Block.** The model performs attention operations conditioned on key factors influencing the regional coastal system, including lead time, boundary conditions, and river inflows. Conditioning on lead time enables the model to predict multiple time intervals with a single model, eliminating the need to train separate models for each lead time, which is particularly beneficial for tasks requiring high temporal resolution. Boundary conditions and river inflows are essential for capturing interactions between the system and its external environment, as these significantly influence the internal processes of the coastal

system. To incorporate these conditions into the ViT architecture, we encode lead time, boundary conditions, and river inflows into embeddings. The overall conditioning vector is obtained by summing the individual embeddings:

$$\mathbf{z}_{\text{con}} = \mathbf{z}_{\text{lead}} + \mathbf{z}_{\text{bc}} + \mathbf{z}_{\text{river}}, \qquad (4)$$

where $\mathbf{z}_{\text{lead}}$, $\mathbf{z}_{\text{bc}}$, and $\mathbf{z}_{\text{river}}$ represent the embeddings for lead time, boundary conditions, and river inflows, respectively. Lead time is transformed into an embedding using a single linear layer. River inflows, represented as features for multiple rivers, are flattened into a single vector and encoded with a linear layer. For boundary conditions, the four boundary lines of the domain are flattened into 1D representations and processed through two 1D convolutional layers followed by a linear layer to extract spatial patterns. These embeddings are then integrated into the attention and feed-forward layers using adaptive layer normalization (adaLN) (Perez et al., 2018), a widely adopted method in generative models (Karras et al., 2019; Peebles & Xie, 2023) and global weather forecasting (Nguyen et al., 2024). The adaLN operation is defined as:

$$\text{AdaLN}(\mathbf{z}, \mathbf{z}_{\text{con}}) = \gamma(\mathbf{z}_{\text{con}}) \cdot \mathbf{z} + \beta(\mathbf{z}_{\text{con}}), \qquad (5)$$

where $\mathbf{z} = \mathbf{z}_{agg} + \mathbf{p}_{phy}$, $\gamma(\mathbf{z}_{\text{con}})$ and $\beta(\mathbf{z}_{\text{con}})$ are scale and shift parameters dynamically generated from the conditioning vector $\mathbf{z}_{\text{con}}$ with linear projection. This dynamic conditioning framework enables the integration of spatial, temporal, and physical context, enriching the model's representation of coastal dynamics.

## 4.2. Training Strategy

Accurately predicting coastal ocean variables for arbitrary lead times presents several challenges. Coastal dynamics are inherently complex, requiring high temporal resolution to capture subtle physical interactions. Additionally, long-lead forecasts demand a model capable of maintaining stability over iterative predictions.

Predicting absolute states $\mathbf{X}(t_0 + \Delta t)$ can be particularly difficult when the changes between the initial condition $\mathbf{X}(t_0)$ and the lead time condition are relatively small. Therefore, we adopt a training strategy (Nguyen et al., 2024; 2023) that predicts the temporal difference $\Delta\mathbf{X} = \mathbf{X}(t_0 + \Delta t) - \mathbf{X}(t_0)$. Moreover, to enhance generalization, we randomly sample $\Delta t$ from a predefined set of time intervals during training. This randomized forecasting strategy provides two key advantages: (1) it effectively augments the training data by exposing the model to a diverse range of temporal intervals, (2) it enables a single trained model to handle predictions for arbitrary time intervals during inference, eliminating the need to train multiple models for different intervals, and (3) it allows the model to efficiently predict long lead times by leveraging large intervals while maintaining high temporal resolution using smaller intervals.

For long-lead forecasts, we adopt an autoregressive approach where the model predicts $\widehat{\Delta\mathbf{X}}$ iteratively and feeds its output back as input for subsequent steps. However, this iterative strategy introduces cumulative errors over extended lead times. To handle this, we employ an autoregressive training objective (Kurth et al., 2023; Lam et al., 2023) that incorporates multiple rollout steps during training. Specifically, the model is rolled out over $K$ steps during training, and the loss is averaged across these steps to reduce error accumulation:

$$\mathcal{L} = \mathbb{E}\left[\frac{1}{KCHW}\sum_{k=1}^{K}\sum_{c=1}^{C}\sum_{i=1}^{H}\sum_{j=1}^{W}\left(\widehat{\Delta\mathbf{X}}_k^{cij} - \Delta\mathbf{X}_k^{cij}\right)^2\right],$$
$$(6)$$

where $K$ is the number of rollout steps, $C$ is the number of variables, and $H$ and $W$ are the horizontal dimensions. Our training procedure is divided into two stages. In the first stage, the model is trained to perform one-step predictions. In the second stage, the model is fine-tuned using the multistep loss, where the same sampled $\Delta t$ is used for all steps within each rollout. This two-stage training strategy ensures that the model achieves both short-term accuracy and long-term stability, making it capable of effectively emulating coastal dynamics over various lead times.

## 4.3. Inference

For inference, the deep learning model is used iteratively, with each forecasted result serving as the input for the next step. To reduce cumulative forecast errors, we adopt the greedy algorithm proposed in (Bi et al., 2023), which selects the largest available lead time model that fits within the remaining prediction horizon. This approach can reduce the number of iterations needed for long-lead forecasts. For instance, with a model capable of predicting intervals of 1, 3, 6, and 12 hours, a 26-hour forecast would require two 12-hour predictions followed by two 1-hour predictions.

## 5. Evaluation

### 5.1. Experimental Setup

We split the dataset into training, validation, and test sets chronologically. Specifically, the first eight years of data are used for training, while the 9th and 10th years are used for validation and testing, respectively. The model follows the ViT-Base (Dosovitskiy et al., 2020) configuration with a hidden dimension of 768, depth of 12, and 12 attention heads, using a patch size of 4. Evaluation metrics include Root Mean Squared Error (RMSE), Mean Absolute Error (MAE), and Pearson Correlation Coefficient (r) to assess prediction accuracy and correlation with ground truth. We evaluate key ocean variables: current velocity $(u, v, w)$, water temperature $(temp)$, salinity $(salt, AKs)$, and free surface elevation

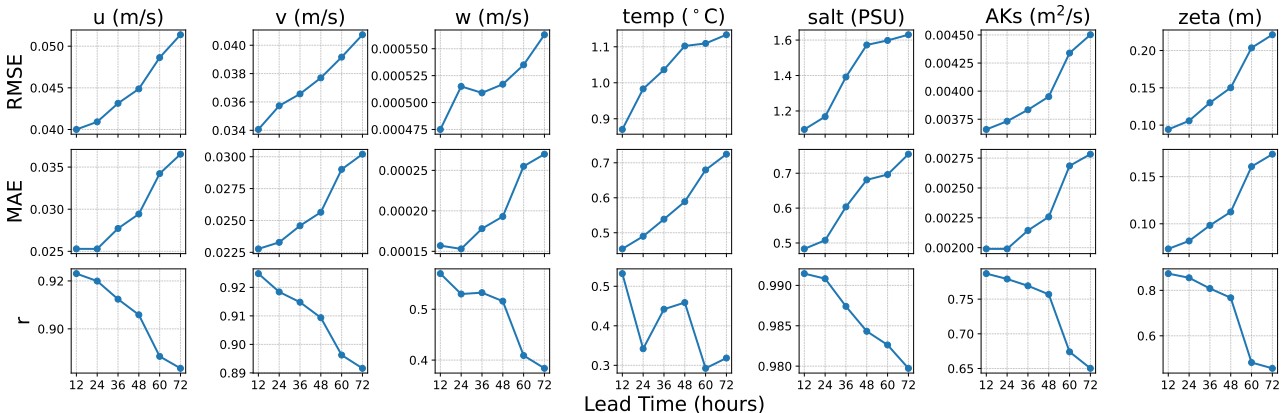

*Figure 3.* Overall performance for long-term forecasting (72-hour lead time) with a 12-hour prediction interval.

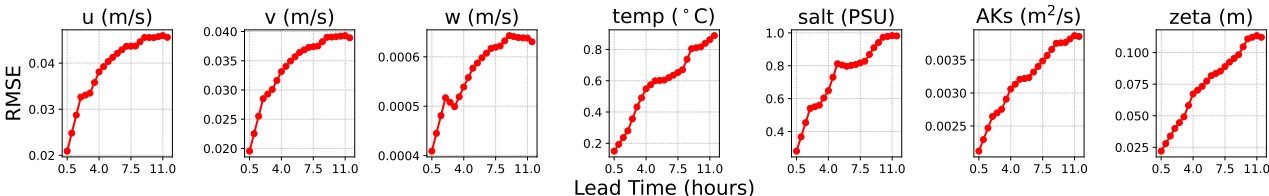

*Figure 4.* Overall performance for short-term forecasting (12-hour lead time) with a 0.5-hour prediction interval.

($\zeta$). Training is conducted on $8 \times$ NVIDIA A100 80GB GPUs using PyTorch's Distributed Data Parallel framework, with a per-GPU batch size of 1. The training process consists of two stages: initial training with one-step prediction, followed by fine-tuning with $K = 4$ autoregressive steps for improved long-term stability. Lead times are sampled from $\{0.5, 3, 12\}$ hours to enhance generalization across different temporal scales.

### 5.2. Overall Model Performance

Figure 3 and 4 show the model's performance for long-term (72-hour) forecasting with a 12-hour interval and short-term (12-hour) forecasting with a 0.5-hour resolution. Due to page limits, we provide only RMSE results for short-term forecasting (See Appendix C for more results). In both cases, performance declines with increasing lead time due to accumulated errors. Anomalies appear in the vertical velocity component ($w$) at 24-hour and 2-hour lead times, where RMSE and MAE deviate from expected trends. This likely results from $w$ being nonzero in only a small fraction of grid cells, making it difficult for the model to capture meaningful patterns and leading to low $r$ values. Similarly, temperature ($temp$) shows low RMSE and MAE but unstable Pearson correlation values, likely due to its low spatial variance, where small absolute errors cause significant correlation fluctuations. For other variables, errors increase

smoothly while correlation decreases with lead time. Overall, the model effectively captures key spatiotemporal patterns but struggles with rare events ($w$) and low-variance fields ($temp$), highlighting challenges in modeling complex coastal dynamics.

**Computational time costs:** We also conducted computational experiments on the inference time of the proposed ViT model. Results show that our ViT model reduces the runtime of ROMS for a 72-hour forecast from 2,477 seconds (with 512 CPU cores on AMD EPYC 7742 64-Core Processors) to 34.14 seconds (on a single A100 GPU), achieving over a 70× speedup.

### 5.3. Ablation Study

To assess the impact of different inputs beyond simulated ocean state variables, we conduct an ablation study by removing (a) meteorological forcings, (b) static physical features, (c) boundary conditions, and (d) river inflow (Figure 5). Excluding meteorological forcings (Figure 5a) significantly increases RMSE for salinity and temperature at longer lead times, confirming their role in capturing atmospheric-driven variability. Static physical features (Figure 5b) primarily affect temperature and salinity, reinforcing their importance in constraining terrain-dependent circulation. Removing boundary conditions (Figure 5c) leads to sharp RMSE increases in velocity, emphasizing their

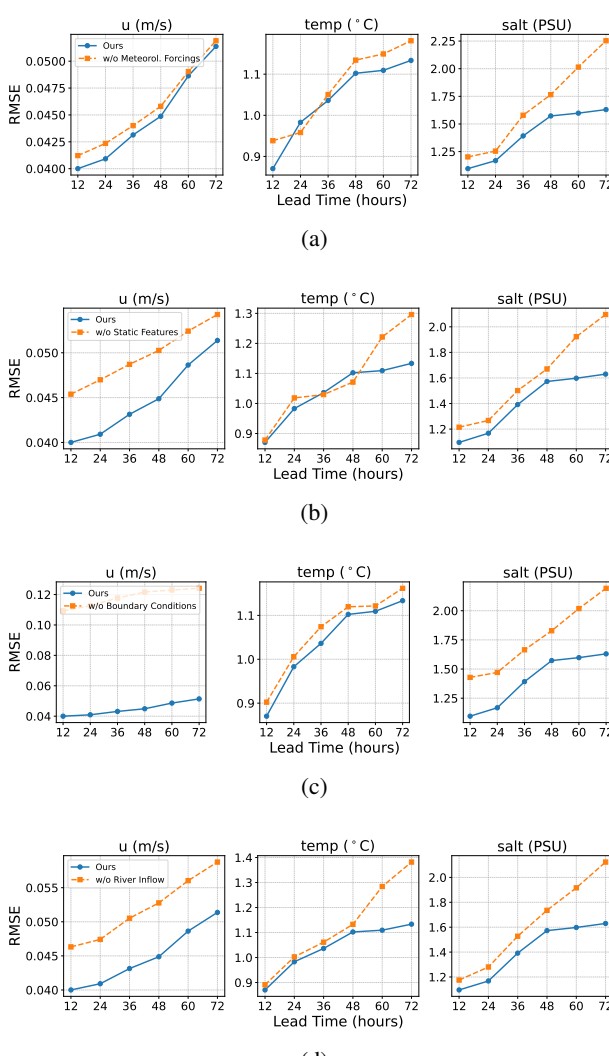

*Figure 5.* Ablation study analyzing the impact of different input variables. (a) Without meteorological forcings, (b) without static physical features, (c) without boundary conditions, and (d) without river inflow.

role in advection processes. Finally, excluding river inflow (Figure 5d) most strongly affects salinity, with a noticeable impact on temperature and a smaller but evident degradation in velocity.

### 5.4. Scaling Test

To assess the effect of model scaling, we compare three ViT variants: ViT-Tiny (ViT-T), ViT-Small (ViT-S), and ViT-Base (ViT-B), differing in hidden dimension, attention heads, and depth (see Appendix B for details). Figure 6 shows RMSE across key ocean variables, where ViT-B consistently outperforms ViT-T and ViT-S. The performance gap widens at longer lead times (48–72 hours), indicating that larger models improve predictive accuracy, especially

for complex processes with a large-scale dataset.

### 5.5. Case Study

To qualitatively evaluate performance, we conduct a case study using January 2, 2016, as the initial condition, generating forecasts for 1, 12, and 24-hour lead times. Figure 7 visualizes temperature and salinity predictions, where rows represent lead times and columns show ground truth (Label), model output (Prediction), and the difference between them (Error, computed as Label – Prediction). At 1 hour, the model accurately captures spatial patterns with minimal error, especially in dynamic regions. At 12 hours, deviations emerge, particularly near coastal boundaries and estuarine inlets due to accumulated errors. By 24 hours, errors become more pronounced, highlighting the challenge of long-term forecasting.

## 6. Discussions and Future Works

This paper introduces *CoastalBench*, a decade-long high-resolution dataset for modeling regional coastal processes, along with a Vision Transformer (ViT)-based approach for emulating these complex dynamics. Through extensive evaluations, we demonstrate that our model achieves competitive performance. The ablation study highlights the importance of meteorological forcings, boundary conditions, and river inflows in improving prediction accuracy, while the scaling test confirms that increasing model capacity enhances predictive performance.

Despite these promising results, several limitations remain. First, error accumulation over long-term predictions remains a challenge, particularly in highly dynamic regions such as estuarine zones and coastal boundaries. Second, due to computational constraints, we are unable to experiment with larger models such as ViT-Large or ViT-Huge, nor fine-tune the model with longer autoregressive steps, which may weaken its long-term forecasting capability. Third, our current deep learning model is relatively simple and lacks architectural modifications specifically designed to handle inputs like external forcings and boundary conditions effectively. Finally, the model is purely data-driven and does not incorporate physical constraints, which may lead to physically inconsistent predictions in certain cases.

For future work, we plan to expand the dataset to cover additional coastal regions and incorporate data assimilation techniques to enhance realism. Another potential extension of the dataset is to provide ensemble simulations for uncertainty quantification. On the modeling side, we aim to explore larger-scale model to assess potential performance gains, as well as develop specialized modules to better handle the unique inputs of regional coastal models.

By releasing *CoastalBench*, we aim to provide a standard-

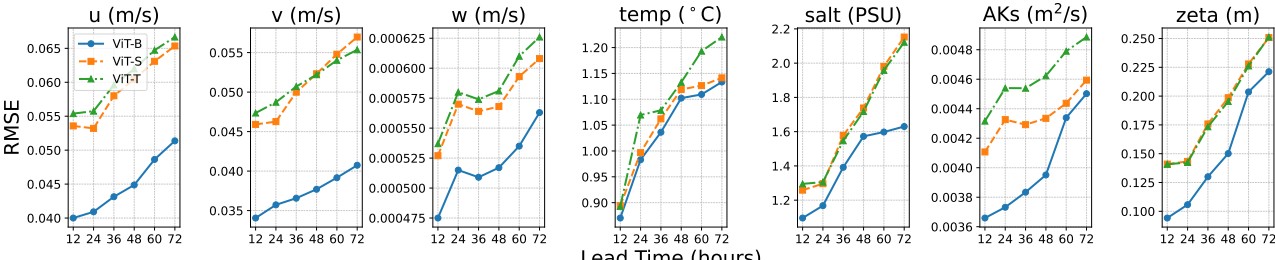

*Figure 6.* Effect of model scaling on performance.

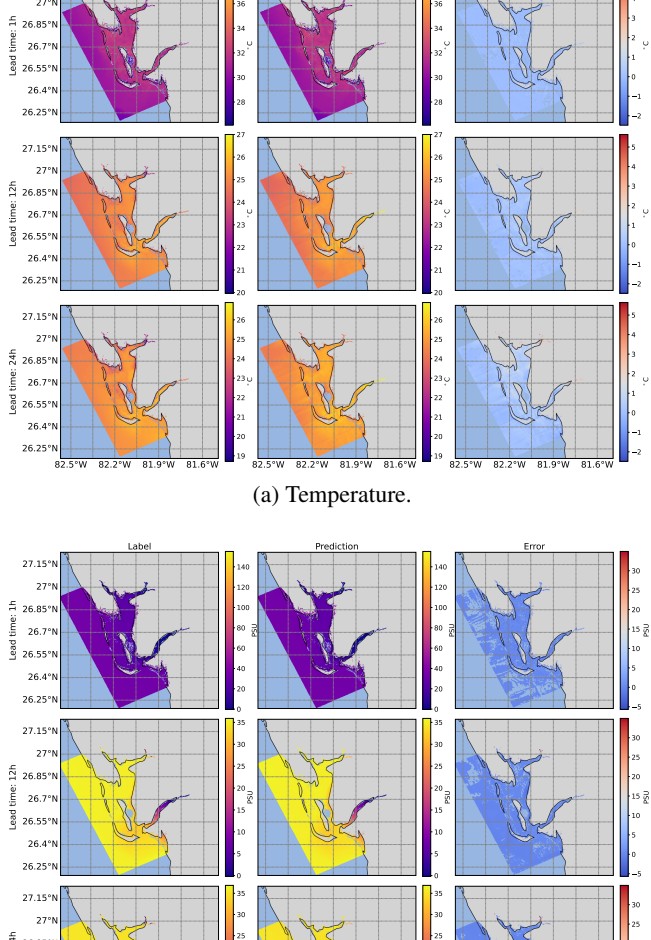

(a) Temperature.

(b) Salinity.

*Figure 7.* Case study visualization of model predictions. (a) Temperature and (b) salinity forecasts for 1, 12, and 24-hour lead times. Each row corresponds to a different lead time, while columns represent ground truth (Label), model prediction (Prediction), and the difference between them (Error, computed as Label – Prediction).

ized benchmark that helps further advancements in deep learning for coastal modeling, encouraging the broader research community to address the identified challenges.

## Acknowledgments

This material is based upon work supported by the National Science Foundation (NSF) under Grant No. IIS-2147908, IIS-2207072, OAC-2152085, OAC-2402946, and OAC-2410884.

## Impact Statement

This work advances the integration of deep learning into coastal ocean modeling by introducing a high-resolution simulation dataset tailored for surrogate modeling. Our dataset enables the development of more accurate and scalable machine learning models for forecasting coastal dynamics. These advances have the potential to support critical applications such as storm surge prediction, water quality monitoring, and climate adaptation planning—ultimately contributing to the resilience of coastal communities and economies.

Nonetheless, as with all data-driven approaches, deep learning-based models remain sensitive to biases in training data and may exhibit uncertainties that impact their reliability in high-stakes scenarios such as disaster response. To address these limitations, rigorous validation and thoughtful integration with physics-based numerical models are essential. Furthermore, the computational demands of training such models may pose barriers to accessibility, underscoring the importance of open-source tools and datasets to democratize progress in this field.

Overall, this work contributes to the broader landscape of AI for scientific discovery, highlighting both the opportunities and responsibilities involved in applying machine learning to Earth system modeling. Ensuring robust performance, transparent evaluation, and collaboration with domain experts will be critical to translating these advances into meaningful societal impact.

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

# A. Variables in *CoastalBench*

| Category | Abbreviation | Description | Unit |
|---|---|---|---|
| Ocean | $u$ | Horizontal velocity (east-west) | m/s |
| | $v$ | Horizontal velocity (north-south) | m/s |
| | $w$ | Vertical velocity | m/s |
| | $temp$ | Water temperature | °C |
| | $salt$ | Water salinity | PSU |
| | $AKs$ | Salinity vertical diffusion coefficient | $m^2/s$ |
| | $\zeta$ | Free surface elevation | m |
| Meteorological | $Pair$ | Surface air pressure | mbar |
| | $Tair$ | Surface air temperature | °C |
| | $Qair$ | Surface air relative humidity | % |
| | $Rain$ | Rainfall rate | $kg/m^2s$ |
| | $swrad$ | Sun's shortwave radiation | $W/m^2$ |
| | $lwrad$ | Sun's longwave radiation | $W/m^2$ |
| | $Uwind$ | Surface u-wind component | m/s |
| | $Vwind$ | Surface v-wind component | m/s |
| River | $river\_transport$ | River runoff mass transport vertical profile | Scalar |
| | $river\_temp$ | River runoff potential temperature | °C |
| | $river\_salt$ | River runoff salinity | PSU |
| Static | $h$ | Bathymetry | m |
| | $f$ | Coriolis parameter | $s^{-1}$ |
| | $pm$ | Curvilinear coordinate metric | $m^{-1}$ |
| | $pn$ | Curvilinear coordinate metric | $m^{-1}$ |

*Table 4.* Details of variables.

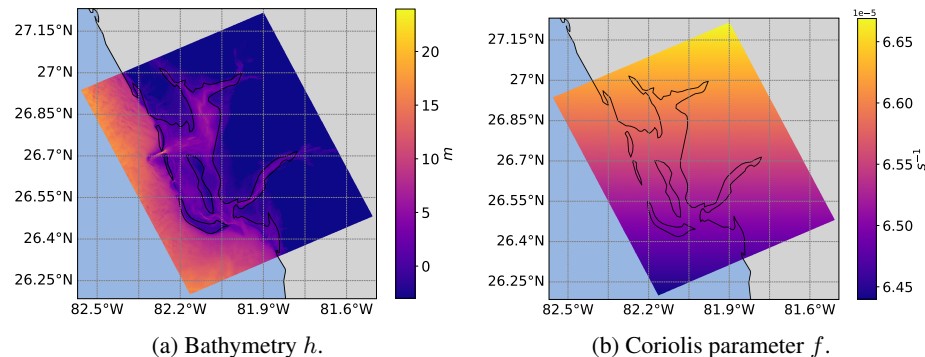

(a) Bathymetry $h$.   (b) Coriolis parameter $f$.

*Figure 8.* Visualization of two static features.

As shown in Table 4, *CoastalBench* includes key ocean state variables including current velocity components ($u, v, w$ for two horizontal and vertical directions), temperature ($temp$), salinity ($salt$), salinity vertical diffusion coefficient ($AKs$), and free-surface elevation ($\zeta$), providing a detailed view of the evolving state of the coastal system. Additionally, external forcings such as meteorological data including air pressure ($Pair$), air temperature ($Tair$), air heat ($Qair$), precipitation ($Rain$), sun's short ($swrad$), and long wave radiation ($lwrad$), and river inflows are included to account for atmospheric and hydrological drivers that shape the spatiotemporal variability of the region. Static physical features, as shown in Figure 8 including bathymetry ($h$), Coriolis parameter ($f$), and curvilinear coordinate metric ($pm$ and $pn$), are also incorporated, capturing the fixed geographical properties critical to coastal processes. Together, these components provide a comprehensive view of the coastal environment, ensuring the dataset's utility for modeling the complex dynamic processes of the system.

## B. ViT Model Configurations

To assess the impact of model scale, we conducted experiments using three ViT variants: ViT-Tiny, ViT-Small, and ViT-Base. We include the configurations for all three models in Table 5. All models use a patch size of 4 and maintain the same input and output.

| Model | Hidden Dimension | Depth (Layers) | Attention Heads |
|-------|------------------|----------------|-----------------|
| ViT-Tiny | 192 | 12 | 3 |
| ViT-Small | 384 | 12 | 6 |
| ViT-Base | 768 | 12 | 12 |

*Table 5.* ViT model configurations used in the scaling test.

## C. Extra Results

Figure 9 and 10 show the results of short-term (12-hour) forecasting with a 0.5-hour resolution on MAE and Pearson Correlation Coefficient, where performance declines with increasing lead time due to accumulated errors.

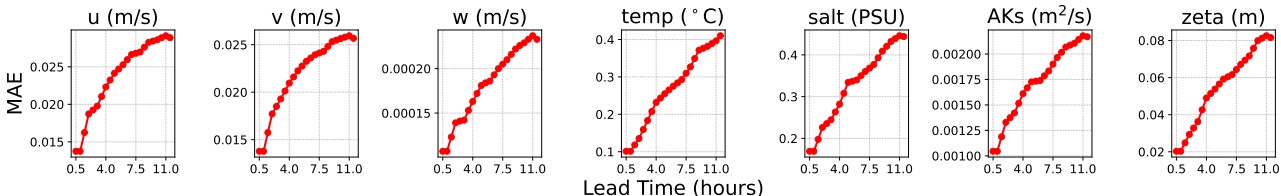

*Figure 9.* Overall performance for short-term forecasting (12-hour lead time) with a 0.5-hour prediction interval (MAE).

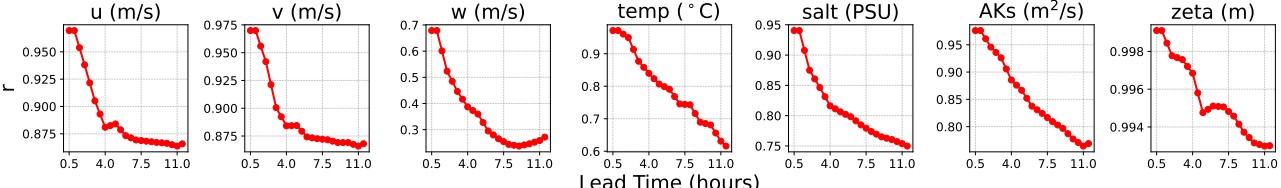

*Figure 10.* Overall performance for short-term forecasting (12-hour lead time) with a 0.5-hour prediction interval (r).

