# OpenReview forum: "CoastalBench: A Decade-Long High-Resolution Dataset to Emulate Complex Coastal Processes"
_ICML.cc/2025/Conference — ICML 2025 poster_

### Official Review · Reviewer_NvM4 · 2025-02-26

**Overall Recommendation:** 3

**Summary:**

This paper mainly focuses on the dataset construction of simulating coastal processes via the Regional Ocean Modeling System (ROMS), considering ocean, meteorological, river and static variables. The work builds a ViT-based (Vision Transformer) network to use the dataset for coastal ocean variable prediction.

## update after rebuttal
Thanks for the responses about my question, and I upgrade my rating. I also suggest to provide the comprehensive literature with the analyses and comparisons to the proposed method, if it's finally accepted.

**Claims And Evidence:**

The dataset is evaluated by customizing a ViT model, but more related models are not considered for comparison, thus the claims may not be convincing enough.

**Essential References Not Discussed:**

The oceanic and atmospheric variable prediction as well as the related datasets should be considered for discussion and comparison.

**Experimental Designs Or Analyses:**

The experiments and analyses are conducted on the constructed dataset and the customized ViT-based model, but there is no comparison, thus the evaluation is insufficient to validate the contributions of this work.

**Methods And Evaluation Criteria:**

The evaluation is only conducted with the given ViT-based model without considering more related works about the oceanic and atmospheric variable prediction, so it's not convincing enough to evaluate the contributions of the dataset and model.

**Other Comments Or Suggestions:**

It's better to reconsider the work with comprehensive literature review.

**Other Strengths And Weaknesses:**

- The contributions are not well reflected in the current version.
- The originality is about the dataset but there is no comprehensive validation on it, thus the significance is not convincing.
- For the dataset, it's better to be validated via the other popular models for the correctness and soundness.

**Questions For Authors:**

What are the major differences compared to similar related works from the perspective of contributions?

**Relation To Broader Scientific Literature:**

It's helpful to build the dataset for oceanic and atmospheric variable prediction.

**Theoretical Claims:**

This is not available for this work.

---

> ### Author Rebuttal · Authors · 2025-04-01
>
> We thank the reviewer for the thoughtful feedback and hope our responses clarify the concerns.
>
> **Related Models:** Thank you for raising this important point. We agree that baselines are important. To our knowledge, no existing deep learning method is specifically designed for complex regional coastal processes, so we plan to include a 3D U-Net as a baseline in the final version. Due to the large scale of our dataset, we were unable to complete this comparison for the rebuttal. If there are specific baselines you recommend, we welcome your suggestions and will aim to include them.
>
> **Related Datasets:** We include a comparison table of representative works on regional coastal ocean modeling, highlighting that existing datasets are generally smaller in scale and focus on simpler processes with limited variable coverage.
> | Dataset Source & Reference                     | Region/Domain          | Spatial Resolution     | Temporal Resolution | Time Span               | Variables                             |
> |------------------------------------------------|------------------------|------------------------|---------------------|--------------------------|----------------------------------------|
> | Kumar & Leonardi (2023)                  | Morecambe Bay, UK      | ~1 km             | Hourly              | 500 storm simulations     | Wave height, depth, sediment           |
> | Ishida et al. (2020)| Japan          | 0.25° (~25 km)         | 1 hour              | 40 years (1979–2019)     | Wind speed, pressure             |
> | Melo et al. (2021)                        | Idealized XBeach | ~1–10 m                | 10 min              | 7 days                   | Bed level, flow, sediment              |
> | Wei et al. (2021)| South China Sea        | ~1 km                  | Hourly              | 1 year       | Wave fields, wind speed                      |
> | O'Donncha et al. (2019)| U.S. West Coast | ~1–2 km           | Hourly              | 1 year                   | Wave fields, wind speed                     |
> | **CoastalBench (Ours)** | Charlotte Harbor, USA  | ~100 m | 30 min            | 10 years (2008–2017)     | Wave fields, temperature, salinity, air pressure, temperature, humidity, rainfall, sun radiation, wind speed, etc. |
>
> Kumar, P., & Leonardi, N. (2023). A novel framework for the evaluation of coastal protection schemes through integration of numerical modelling and artificial intelligence into the Sand Engine App.
>
> Ishida, K., et al. (2020). Hourly-scale coastal sea level modeling in a changing climate using long short-term memory neural network.
>
> Melo, C. B., et al. (2023). Coastal morphodynamic emulator for early warning short-term forecasts.
>
> Wei, Z., et al. (2022). A convolutional neural network based model to predict nearshore waves and hydrodynamics.
>
> O'Donncha, F., et al. (2019). Ensemble model aggregation using a computationally lightweight machine-learning model to forecast ocean waves.

---

### Official Review · Reviewer_faRw · 2025-03-09

**Overall Recommendation:** 3

**Summary:**

This paper provides a large-scale, high-resolution coastal simulation dataset to train and evaluate deep learning models. The dataset contains various oceanography variables alongside external atmospheric and river forcings. Then, the author proposes a customized ViT model that takes initial and boundary conditions and external forcings as input and predicts ocean variables at varying lead times. The model achieves competitive performance.

**Claims And Evidence:**

The claims made in the submission are not all supported by evidence.

1. The authors claim existing studies focus on small datasets and simple processes. They should include a thorough comparison with pervious datasets, highlighting their contributions and how their dataset is unique from others (preferably in a tabular format).

2. This paper propose a physics-aware positional embedding, which is uncommon. Experiments are needed to prove the validity of this design.

**Essential References Not Discussed:**

Essential references are included.

**Experimental Designs Or Analyses:**

I checked the soundness/validity of experimental designs. The main problem is that baselines with numerical methods and existing deep learning methods on your dataset are missing, making it unclearly how well the proposed method is.

**Methods And Evaluation Criteria:**

The proposed methods and evaluation criteria make sense for the application.

**Other Comments Or Suggestions:**

1. A small typo on Page4 Line193 left-side: "which **an** be used"

2. Descriptions after Equation 5 (Page 4) are hard to follow, due to some notation missing or abusing.

3. On Page 3, right side, under "Dataset Construction," all citations are missing. These references appear in the Supp. but are missing from the main text.

**Other Strengths And Weaknesses:**

Strengths:
1. Large-scale dataset solving the problem of lack of large-scale public dataset

Weakness:
1. The description of the dataset is not clear or comprehensive. Apart from Table 1, a dataset summary table detailing the dataset volume, data shape, format, number of variables, etc. is missing.

2. Key experimental results are missing. Moreover, a more comprehensive analysis of both the dataset and the proposed framework will be helpful.

3. The writing and organization of the paper should be improved.

**Questions For Authors:**

Both as part of External Forcings, why does your deep learning model treat Meteorological Forcings and River Inflow differently? Will it significantly influence the model performance if the input Meteorological Forcings are used as an additional condition?

**Relation To Broader Scientific Literature:**

The dataset is constructed via existing simulation method ROMS (Shchepetkin & McWilliams, 2005), and some variables (e.g., atmospheric forcings were obtained from the North American Regional Reanalysis (NARR) (Mesinger et al., 2006).

**Theoretical Claims:**

There is no proof of theoretical claims in this submission.

---

> ### Author Rebuttal · Authors · 2025-04-01
>
> Thank you for your comments and thoughtful questions. Below are our responses:
>
> **Comparison with existing datasets:** We include a comparison table of representative works on regional coastal ocean modeling, highlighting that existing datasets are generally smaller in scale and focus on simpler processes with limited variable coverage.
> | Dataset Source & Reference                     | Region/Domain          | Spatial Resolution     | Temporal Resolution | Time Span               | Variables                             |
> |------------------------------------------------|------------------------|------------------------|---------------------|--------------------------|----------------------------------------|
> | Kumar & Leonardi (2023)                  | Morecambe Bay, UK      | ~1 km             | Hourly              | 500 storm simulations     | Wave height, depth, sediment           |
> | Ishida et al. (2020)| Japan          | 0.25° (~25 km)         | 1 hour              | 40 years (1979–2019)     | Wind speed, pressure             |
> | Melo et al. (2021)                        | Idealized XBeach | ~1–10 m                | 10 min              | 7 days                   | Bed level, flow, sediment              |
> | Wei et al. (2021)| South China Sea        | ~1 km                  | Hourly              | 1 year       | Wave fields, wind speed                      |
> | O'Donncha et al. (2019)| U.S. West Coast | ~1–2 km           | Hourly              | 1 year                   | Wave fields, wind speed                     |
> | **CoastalBench (Ours)** | Charlotte Harbor, USA  | ~100 m | 30 min            | 10 years (2008–2017)     | Wave fields, temperature, salinity, air pressure, temperature, humidity, rainfall, sun radiation, wind speed, etc. |
>
> Kumar, P., & Leonardi, N. (2023). A novel framework for the evaluation of coastal protection schemes through integration of numerical modelling and artificial intelligence into the Sand Engine App.
>
> Ishida, K., et al. (2020). Hourly-scale coastal sea level modeling in a changing climate using long short-term memory neural network.
>
> Melo, C. B., et al. (2023). Coastal morphodynamic emulator for early warning short-term forecasts.
>
> Wei, Z., et al. (2022). A convolutional neural network based model to predict nearshore waves and hydrodynamics.
>
> O'Donncha, F., et al. (2019). Ensemble model aggregation using a computationally lightweight machine-learning model to forecast ocean waves.
>
>
> **Baselines:** Thank you for raising this important point. We agree that baselines are important. To our knowledge, no existing deep learning method is specifically designed for complex regional coastal processes, so we plan to include a 3D U-Net as a baseline in the final version. Due to the large scale of our dataset, we were unable to complete this comparison for the rebuttal. If there are specific baselines you recommend, we welcome your suggestions and will aim to include them.
>
> **Physics-aware positional embedding:** We have provided the ablation study to show the effectiveness of incorporating the physics information into positional embedding, as shown in Figure 5(b).
>
> **Dataset summary table:** Thank you for the suggestion. We added the following dataset summary table to clearly describe key properties of the dataset:
>
> | Property                  | Description|
> |--|--------|
> | **Temporal Coverage**     | 2008–2017 (10 years)|
> | **Temporal Resolution**   | 30 minutes|
> | **Spatial Domain**        | Charlotte Harbor, Florida, USA (~800 km²)|
> | **Grid Type**             | Non-uniform 3D curvilinear mesh|
> | **Grid Dimensions**       | 898 (lat) × 598 (lon) × 12 (vertical levels)|
> | **Average Horizontal Resolution** | ~120 m × 100 m|
> | **Number of Variables**   | 22 (See Table 2 for details)|
> | **Data Format**           | NetCDF|
> | **Total Volume**          | ~18 TB|
> | **Simulation Model**      | Regional Ocean Modeling System (ROMS)|
>
>
> **Other comments or suggestions:** Thanks for your suggestions. We have corrected the typo on Page 4, clarified the notations and descriptions following Equation (5), and added the missing citations in the “Dataset Construction” section on Page 3.

---

> > ### Comment · Reviewer_faRw · 2025-04-03
> >
> > Thank you for the responses. After reviewing your answers and considering other reviewers’ comments, I keep my original score.

---

### Official Review · Reviewer_oeQD · 2025-03-11

**Overall Recommendation:** 3

**Summary:**

This paper introduces a decade-long, high-resolution dataset for modeling complex coastal processes in the area of Charlotte Harbor, Florida, USA.  The dataset is generated using a validated numerical model, ROMS. A flexible ViT model is designed to ingest a multitude of diverse data sources (e.g. initial, boundary, external and static conditions, river inflows, etc.) to predict ocean variables at various lead times. The ablation study reveals the importance of incorporating each of these data sources to improve prediction accuracy.

**Claims And Evidence:**

- The dataset seems to be carefully designed and validated. The ablations clearly show that incorporating all data sources is important for optimal predictability. The network architecture is well motivated and contextualised to prior work.

**Essential References Not Discussed:**

n/a

**Experimental Designs Or Analyses:**

Except for my aforementioned concerns/questions on the experimental setup (see the Methods and Evaluation Criteria section), the training and evaluation metrics/analysis seem sound to me.

**Methods And Evaluation Criteria:**

I'm missing more discussion on the problem setting/experiments tackled in the paper. Right now, the created dataset is interesting, but the experimentations and evaluation criteria lack motivation.
- What motivates this problem setting? Why is it a good problem to tackle with deep learning? What are some specific downstream applications of your model? The impact statement reads well, but I'd like to better understand how it links to the specifics (i.e., data, problem setup, evaluation) in this paper.
- Why can't you simply use ROMS for the same problem (or what's the issue with that)? Is the goal to essentially emulate ROMS? If so, is computational speed the main reason? If yes, please include a runtime benchmark.
- Why not include observational data in the dataset? E.g. RECON, which was already used to validate ROMS (see appendix); are there others? How exactly was ROMS validated against observations? How accurate are its simulations? If the reliance on ROMS simulations is a limitation (it seems to me), can you explicitly mention it in the main text?
- The data only covers the region of Charlotte Harbor, Florida, USA. How complicated/feasible is it to expand it to other regions (or provide the necessary tools to users to expand it themselves to regions they're interested in)?
- Is the current model, or its errors, sufficiently good already to be useful in practice? If not, which evaluation procedures/scores could enlighten potential users of the benchmark when this is the case? Is there any uncertainty in the data, i.e. would probabilistic models/metrics be potentially well-suited for this benchmark?
- I don't understand why the boundary conditions, meteorological forcings, and river inflow inputs are all from the "future" (i.e. the same timestep, $t_0+\Delta t$, for which you aim to predict the coastal ocean variables). How can this be useful in practice? Won't this make it impossible to perform real-world predictions (since you'd need to wait for the input data at time $t_0+\Delta t$ before running your model)?
- Is the 8:1:1 train/val/test split random? For temporal prediction tasks, it's recommended to split by time.

**Other Comments Or Suggestions:**

n/a

****** After rebuttal: Updated score from 2 -> 3

**Other Strengths And Weaknesses:**

Strengths:

The dataset structure is quite interesting, including various data sources and dimensionalities. It makes it an interesting problem to design an appropriate architecture that can ingest all these sources. The authors introduce an adaptation of ViT that's suitable for this data. Its design, especially for how to condition the model on various forcings, is very well motivated and makes sense. The design is quite general and could be useful for different problems/datasets. The ablations show clearly that incorporating all these forcings boosts performance.

Weaknesses:

1. See the Methods And Evaluation Criteria section.
2. There's little practical information on the dataset that would be important for potential users. (How) Does it adhere to FAIR principles? What format do you use? Was any postprocessing done? Where will the data be hosted? Given the size/diversity of the data, did you take any steps to make it easier for ML practitioners to download it and easily get a simple model running? I'd recommend the authors to take a look at the NeurIPS call for datasets and benchmarks and ensure that such key pieces of information (for a benchmark dataset) are included in the paper/supplementary.

**Questions For Authors:**

- What do you mean by *"we plan to expand the dataset to cover additional coastal regions and incorporate data assimilation techniques to enhance realism"*? How is the current data lacking realism? What's missing?
- Why does the colorbar in your absolute error plots (Fig. 7, right column) start at negative values? Can you fix this?

**Relation To Broader Scientific Literature:**

I am unfamiliar with the literature on coastal circulations/ecosystems, so this is hard for me to judge. From the perspective of the dataset itself and its structure, I think that these are potentially interesting to a broader community (see the Strengths section).

**Theoretical Claims:**

There are no such claims.

---

> ### Author Rebuttal · Authors · 2025-04-01
>
> Thank you very much for your valuable review; it is crucial for improving the quality of our manuscript.
>
> **Motivation and applications:** Our problem setting is motivated by the need to efficiently emulate complex coastal processes for practical applications. High-resolution numerical models such as ROMS are computationally expensive, while deep learning offers a fast and scalable alternative. The dataset includes variables critical for key downstream tasks: for example, storm surge and coastal flood forecasting benefit from predictions of free surface elevation; water quality and stratification modeling rely on temperature and salinity; and sediment transport analysis depends on vertical diffusivity.
>
> **Why not use ROMS:** The goal is to emulate ROMS, with the primary motivation being computational efficiency—numerical models like ROMS are significantly slower than deep learning. For example, our experiments show that the proposed model reduces the runtime of ROMS for a 72-hour forecast from 2,477 seconds (using 512 CPU cores) to 34.14 seconds on a single A100 GPU, achieving over a 70× speedup. We have added a detailed runtime benchmark to our manuscript.
>
> **Use of observational data:** We recognize the value of observational data, but they are often limited in spatial and temporal coverage. For instance, RECON provides measurements at sparse, fixed locations and irregular intervals. The ROMS model used in this study was previously validated against RECON observations (Hewageegana et al., 2023), showing strong agreement in key physical processes such as water level variability, currents, salinity, and temperature. This prior validation supports the use of ROMS outputs for training the deep learning model. While our focus is on emulating ROMS rather than evaluating it against observations, we agree that the reliance on ROMS simulations could be considered a limitation. We will clarify this in the main text.
>
> Hewageegana, V. H., Olabarrieta, M., & Gonzalez-Ondina, J. M. (2023). Main Physical Processes Affecting the Residence Times of a Micro-Tidal Estuary. Journal of Marine Science and Engineering, 11(7), 1333.
>
> **Expand data to other regions:** Expanding the dataset to other regions is non-trivial. First, it requires region-specific input data such as boundary conditions and external forcings. Second, running high-resolution ROMS simulations is computationally intensive. Third, manual calibration by oceanographers and domain experts is essential to ensure simulation quality.
>
> **Evaluation procedures:** There is no universally accepted standard for determining whether a coastal ocean model is "sufficiently good" for practical use, as acceptable error thresholds depend on the specific downstream application (e.g., storm surge vs. long-term climatology). In our case, the goal is only to approximate ROMS outputs as closely as possible.
>
> **Uncertainty:** This is a great point. While forecasting inevitably introduces uncertainty, our dataset is based on a ROMS hindcast calibrated using real-world observations. However, we do not currently provide ensemble simulations or uncertainty quantification. Probabilistic models or metrics are not suitable for this scenario.
>
> **Future inputs:** This is a fair concern. In our setup, the inputs at the target time are assumed to be available from external forecast systems (e.g., global atmospheric forecasts) at coarse resolution. This mirrors real-world practice, where future forcings and boundary conditions are obtained from other (global) forecast models, and regional numerical models take them as input. Making predictions using only initial conditions is not reliable for regional systems due to their sensitivity to future external forcings and boundary conditions. This is actually one of the key motivations for proposing this regional dataset, which differs from global forecasting datasets.
>
> **Dataset split:** Our dataset is indeed split chronologically. Specifically, the first 8 years of data are used for training, while the 9th and 10th years are used for validation and testing.
>
> **Practical information:** Thanks for the suggestion. We have added practical dataset details in the paper. The dataset will be hosted on Hugging Face and provided in standard NetCDF (.nc) format. We also include a compressed version in PyTorch (.pth) format (converted from float64 to float16) for efficient use in ML pipelines. The full dataset is approximately 18 TB. To improve usability, we provide a lightweight subset and a base training script to help users quickly run models.
>
> **Q1:** The current dataset uses manually calibrated ROMS simulations based on observations, which helps ensure general adherence to realism. However, we do not apply data assimilation, and the outputs may not perfectly align with real-world observations.
>
> **Q2:** Apologies for the mistake — this is not absolute error, but the difference computed as $Label - Prediction$. We have corrected this in the manuscript.

---

> > ### Comment · Reviewer_oeQD · 2025-04-02
> >
> > Thank you for the clear and satisfying answers. I will raise my score to 3.
> >
> > Those practical dataset details, especially ensuring its accessibility and proper documentation, are really important to get right or this benchmark dataset to be impactful. It's unfortunate that it's not possible to verify how (well) these things have been added to the paper.
> >
> > About the "Future inputs" discussion: This makes the models (including the simulator, ROMS) unusable for real-time monitoring/forecasting, right? If so, this seems like an important limitation to outline in the paper. Please correct me if I'm wrong (e.g., if the latency of the external forecasts is minimal). I understand that using only initial conditions is insufficient, but I'm curious how your emulator would perform when using past- or present-time external information only (i.e., external forecasts for the past/current timesteps only).

---

> > > ### Author Response · Authors · 2025-04-03
> > >
> > > Thank you for your feedback and for raising the score.
> > >
> > > About the "future inputs" concern, our approach is consistent with how state-of-the-art regional models like ROMS are typically used. For example, here are all the regional forecast models operated by the National Oceanic and Atmospheric Administration (NOAA), including several ROMS-based systems, for real-time monitoring and forecasting: https://tidesandcurrents.noaa.gov/models.html
> > >
> > > Because regional models cover a limited domain, they inherently depend on boundary conditions and atmospheric/oceanic forcings from larger-scale forecasting systems. This applies equally to both traditional numerical models and our deep learning emulator. As such, our model can indeed be used for real-time monitoring or forecasting, as long as the necessary inputs (which are usually available from different sources like global forecast systems) are accessible.
> > >
> > > We agree that this dependency on external forecasts introduces latency, which is a common constraint for all regional modeling systems. This further underscores the value of fast models like ours, which can significantly reduce the end-to-end time required for forecasting once inputs become available. We will clarify this further in the paper.

---

### Decision · Program_Chairs · 2025-05-01

**Decision:**

Accept (poster)

**Comment:**

This paper introduces a data set with calculations from regional coastal circulation models. The data set covers a decade-long timescale with 3 temporal resolution of 30 minutes and spatial resolution of about 100 m (on average). The paper also provides preliminary results using a Transformer model. My impression from reading the paper, the reviews and the rebuttal discussions is that the contribution is meaningul and the paper is solid. I think the rebuttal has addressed the main questions by the reviewers and those who have engaged in the discussion have raised the score to acceptance. I largely agree with their assessment and I believe this contribution would be appreciated by part of the ICML audience. For these reasons, I recommend the acceptance of the submissions. Meanwhile, I strongly encourage the authors to incorporate some elements from the rebuttal discussion, such as certain clarifications and answers to questions and the comparison with existing data sets.